# Limited Performance of Machine Learning Models Developed Based on Demographic and Laboratory Data Obtained Before Primary Treatment to Predict Coronary Aneurysms

**DOI:** 10.3390/biomedicines13051073

**Published:** 2025-04-29

**Authors:** Mi-Jin Kim, Gi-Beom Kim, Dongha Yang, Yeon-Jin Jang, Jeong-Jin Yu

**Affiliations:** 1Department of Pediatrics, Asan Medical Center, University of Ulsan College of Medicine, 88, Olympic-ro 43-Gil, Songpa-Gu, Seoul 05505, Republic of Korea; atlranta83@hanmail.net; 2Department of Pediatrics, Seoul National University Children’s Hospital, Seoul 03080, Republic of Korea; ped9526@snu.ac.kr; 3Mega AI Lab Co., Ltd., Seoul 05825, Republic of Korea; dyang@ckt21.com (D.Y.); od00075@amc.seoul.kr (Y.-J.J.)

**Keywords:** Kawasaki disease, machine learning, coronary artery aneurysm

## Abstract

**Background/objectives**: Kawasaki disease is the leading cause of acquired heart disease in children within developed countries. Although treatment with intravenous immunoglobulin (IVIG) significantly reduces the incidence of coronary artery aneurysm (CAA), the risk of it persists, affecting long-term patient outcomes. While intensified primary treatment is recommended for patients at high risk of IVIG resistance or CAA development, a universally accepted predictive model for such resistance remains unestablished. This study aims to develop a machine learning model to predict the occurrence of CAAs prior to initiating IVIG therapy. **Methods**: Data from two nationwide epidemiological surveys conducted between 2012 and 2017 were analyzed, encompassing 17,189 patients with calculable coronary artery z-scores and Harada scores. Various supervised machine learning algorithms were applied to develop a model for predicting CAA. Afterward, unsupervised learning techniques were employed to explore the data’s inherent structure. **Results**: The Harada score’s receiver operating characteristic (ROC) analysis yielded an area under the curve (AUC) of 0.558. The highest AUC among the machine learning models was 0.661, achieved by the Light Gradient Boosting Machine. However, this model’s sensitivity was 0.615, and specificity was 0.647, indicating limited clinical applicability. Unsupervised learning revealed no distinct distribution patterns between patients with/without CAAs. **Conclusions**: Despite utilizing a large dataset to develop a machine learning-based prediction model for CAAs, the performance was unsatisfactory. Future studies should focus on enhancing predictive models by incorporating additional clinical data, such as acute-phase coronary artery diameter measurements, to improve accuracy and clinical utility.

## 1. Introduction

Kawasaki disease is the most common acquired heart disease in children in developed countries [1]. Primary treatment is the administration of 2 g/kg of intravenous immunoglobulin (IVIG), which can reduce the incidence of coronary artery aneurysms based on absolute diameter measurements from 25% to 4% [1]. However, the occurrence of coronary artery complications remains the most important factor determining long-term prognosis.

Intensified primary treatment refers to the concurrent administration of other drugs in addition to immunoglobulin to patients predicted to have a poor prognosis—a resistance to immunoglobulin or the occurrence of coronary artery aneurysm—and is recommended in both the American Heart Association (AHA) and Japanese guidelines [2,3]. However, a globally accepted predictive model for primary immunoglobulin resistance has not yet been established [4], despite reports of successful prediction in Japan [5,6,7]. The AHA clinical guidelines recommend intensified primary treatment for patients at risk of developing coronary artery aneurysm [1,2]. The updated AHA guidelines [2] have recommended the predictive model for coronary artery aneurysm development proposed by Son MBF et al., in which coronary artery dilatation during the acute phase is included as the most important factor [8]. However, as the study by Son MBF et al. includes Asian ethnicity as a risk factor [8], some adjustments to the model may be needed in the context of Republic of Korea, where the entire population is of Asian descent. The most well-known coronary aneurysm occurrence prediction model is the Harada score, which was developed in Japan to predict high-risk patients who require primary immunoglobulin therapy [9]. Since the Harada score was published in 1991, there have been reports on its predictive performance outside of Japan, but its usefulness has been questioned [10,11]. The specificity has been reported to be approximately 51–53%.

Machine learning can provide opportunities to establish models that perform tasks such as pattern recognition, prediction, classification, and clustering across a wide range of human activities [12]. Machine learning can be categorized into supervised learning and unsupervised learning depending on the presence or absence of a dependent variable (in other words, ‘target variable’, ‘response variable’, or ‘label’). Recently, there have been several publications that applied machine learning methods to clinical data of Kawasaki disease [13]. They are summarized as supervised learning with Kawasaki disease diagnosis, immunoglobulin resistance, or coronary aneurysm occurrence as target variables. It is believed that the application of machine learning methods instead of conventional statistical methods was intended to increase the predictive power of the developed model.

This study was designed to develop a machine learning model to predict the occurrence of coronary artery aneurysms before the initiation of immunoglobulin therapy, utilizing a large-scale dataset collected through a nationwide survey. The application of a machine learning model is expected to enhance predictive accuracy. Although measurements of coronary artery diameter during the acute phase are not included in the dataset, it contains essential demographic and laboratory data. In addition, the performance of the developed predictive model can be compared with the Harada score.

## 2. Materials and Methods

### 2.1. Subjects

In two nationwide epidemiological surveys of Kawasaki disease conducted in the Republic of Korea, 14,916 patients were investigated in 2012–2014 and 15,378 patients were investigated in 2015–2017 [14,15]. The data were composed of a combined set of data from two surveys. There were many incomplete data with missing feature values. The final dataset was composed of data from 17,189 patients for whom z score of coronary arteries and Harada score [9] could be calculated. A total of 13,105 patients were excluded, for whom coronary artery diameter measurements were incomplete or for whom the Harada score could not be calculated. Patients in whom only the left circumflex coronary artery was missing in the coronary artery diameter measurements were included.

### 2.2. Data Processing

According to a recent research report [10], the condition for satisfying two constituting variables in calculating the Harada score was modified-the serum C-reactive protein level > 3+ was changed to > 3 mg/dL, and hematocrit < 35% was changed to hemoglobin < 11.6 g/dL (Table 1). If each condition of the seven constituting variables is satisfied, it is considered as 1 point, and if the sum is ≥4, the Harada score is defined as positive.

In two nationwide surveys, maximum coronary artery diameters measured in the early course of disease included the subacute phase [14,15]. However, coronary artery diameters measured during the acute phase, including the pretreatment period, were not investigated. These measurements were used to calculate coronary artery diameter z scores based on published Korean norms [16]. Based on the recommendations of the American Heart Association, coronary artery aneurysm was defined as a z score ≥ 2.5 of any coronary artery [1].

Demographic and laboratory variables, excluding coronary artery diameter measurements, were considered as features in the dataset. Several supervised machine learnings with coronary artery aneurysm as the target variable were performed. In addition, unsupervised machine learning was performed with the set of features to discern the ease of classifying patients with coronary artery aneurysm in the dataset.

Before starting machine learning, we performed some data preprocessing. To ensure data completeness, missing values in the dataset were imputed using the mean strategy. For categorical data, missing values were replaced with default values. To mitigate the impact of extreme values, a three-standard-deviation rule was applied. The mean and standard deviation of each feature were computed, and values exceeding three standard deviations from the mean were capped at the corresponding lower and upper bounds (±3SD). This transformation aimed to reduce the influence of outliers while preserving data integrity.

### 2.3. Supervised Machine Learning

The dataset was split into training and testing sets using a 9:1 ratio. The stratified splitting technique was applied to maintain the class distribution of the target variable across both subsets. Feature scaling was performed to normalize the data and improve model performance. The transformation was fitted on the training set and subsequently applied to both the training and testing sets to prevent data leakage. To address class imbalance in the dataset, the Synthetic Minority Over-sampling Technique (SMOTE) was attempted to generate synthetic samples for the minority class to balance the dataset, but it was ultimately abandoned due to overfitting problems on the training set.

The applied supervised machine learning methods are Logistic Regression, Support Vector Machine, Ensemble method, Random Forest, Gradient Boosting Machine, Light Gradient Boosting Machine, and Multi-Layer Perceptron. Logistic Regression is a statistical method used for binary classification by modeling the probability of a given class. Support Vector Machine constructs a hyperplane in a high-dimensional space to maximize the margin between classes. Ensemble methods, including Random Forest and Explained Boosting Machine, combine multiple models to improve predictive performance and reduce overfitting. Random Forest is a bagging-based ensemble method that constructs multiple decision trees. Gradient Boosting Machine and its variants, such as Light Gradient Boosting Machine, utilize boosting techniques to sequentially improve weak learners, achieving high performance in predictive tasks. Multi-Layer Perceptron is a type of artificial neural network that captures complex patterns through multiple hidden layers. While ensemble and boosting methods focus on combining multiple models for better generalization, neural networks like Multi-Layer Perceptron are designed to learn hierarchical representations of data.

Logistic Regression was performed using two methods: Lasso Regularization and Ridge Regression. There were three hidden layers in the Multi-Layer Perceptron.

### 2.4. Performance Evaluation

The performance evaluation of the prediction model for coronary aneurysms was performed using receiver operating characteristic curve (ROC) analysis. The superiority of the predictive model was evaluated based on the area under the ROC curve (AUC), with a larger area indicating better performance. Additional metrics such as sensitivity, specificity, positive predictive value, negative predictive value, F1-score, and Matthews Correlation Coefficient (MCC) were presented. Five-fold cross-validation (CV) was performed by dividing the training set into five subsets, calculating the ROC AUC for each, and obtaining the mean value. To assess the potential overfitting of the machine learning model, the results from the test set were presented alongside for comparison.

### 2.5. Unsupervised Machine Learning

Principal component analysis was first performed on the set of features. Afterwards, the high-dimensional dataset was reduced to three dimensions using the t-distributed Stochastic Neighbor Embedding (t-SNE) algorithm to facilitate visualization and enhance clustering efficiency. To uncover inherent group structures in the data, an unsupervised clustering approach was applied using the K-Means algorithm. t-SNE is a dimensionality reduction technique that preserves local similarities in high-dimensional data, making it particularly effective for visualizing complex structures. K-Means algorithm is a clustering method that partitions data into K clusters by minimizing intra-cluster variance, providing efficient and interpretable groupings. The optimal number of clusters was set to two, and the algorithm was initialized with a random state to ensure consistency.

SPSS software version 28.0 (IBM Corp., Armonk, NY, USA) was used for the analysis of the Harada score. Python 3.12.4 was used for machine learning, and Spyder IDE 5.5.1 was the package for it.

## 3. Results

The values of demographic and laboratory variables and coronary artery diameter measurements in the entire dataset are presented in Table 2. For variables with missing values, the number of samples for which values were investigated is also displayed.

The results of the Harada score analysis are presented in Figure 1. The AUC was 0.558 (95% confidence interval 0.546–0.569). When the cutoff value of the Harada score for predicting CAA was set to ≥4, the sensitivity was 0.541, the specificity was 0.547, the positive predictive value was 0.208, and the negative predictive value was 0.845.

### 3.1. Supervised Machine Learning

Logistic Regression was performed in two ways, and the results were similar (Table 3). The prediction model with the highest AUC in the ROC analysis was the model using Light Gradient Boosting Machine. The AUC value was 0.661, which seemed to be higher than 0.558 in the Harada score (Figure 1). Across all of the developed prediction models, the mean ROC AUC of the training set obtained from the cross validation results did not seem to differ much from the ROC AUC value obtained from the test set. Therefore, the possibility of overfitting in the training set does not seem to be high. The F1 scores were distributed in the low range of 0.242–0.376, and the MCC was distributed in the low range of 0.100–0.197.

### 3.2. Unsupervised Machine Learning

In the principal component analysis, the cumulative explained variance value for the three features was 0.974 (Figure 2). Therefore, the authors reduced the set of features to three dimensions and split it into two groups (Figure 3a). Finally, individual patients with CAA were marked with different colors to show their distribution within the entire set (Figure 3b). The distribution of patients who have coronary artery aneurysm in the two groups classified according to the three principal components did not seem to show any specific pattern.

## 4. Discussion

In this study, machine-learning-based models demonstrated better performance than the Harada score in ROC analysis. The best machine learning model was the Light Gradient Boosting Machine with an AUC value of 0.661 and a sensitivity of 0.633 and specificity of 0.615. However, these results of metric do not appear to be sufficient performance to be used in clinical settings. Considering that patients with coronary artery aneurysm did not show significant distributional pattern in the two groups divided according to the results of the principal component analysis, it seems that the data of this study were not sufficient for easily classifying patients with coronary artery aneurysm. In this study, statistical methods were not attempted for the development of a predictive model. If a statistical predictive model had been developed, it might have achieved performance comparable to logistic regression-based machine learning. However, it would not have been possible to achieve better performance.

The Harada score originally developed to predict the need for immunoglobulin administration in Kawasaki disease patients [9], has shown varying performance in predicting coronary artery aneurysms across different studies and populations. In a recent study conducted in Iran [10], the sensitivity and specificity of the Harada score were reported as 0.564 and 0.537, respectively, which are similar to the results of the present study. The definition of coronary artery aneurysm was based on the z-score in this study, differing from the definition used when the Harada score was originally developed. This discrepancy is presumed to be one of the factors contributing to the low performance of the Harada score. Further analysis using the original definition of coronary artery aneurysm based on absolute diameter by Japanese Ministry of Health and Welfare [17]—the prediction target of the Harada score—was not performed, as current major guidelines consistently endorse the use of z score–based criteria [1,18]. And additional criteria for defining a coronary artery aneurysm—a diameter 1.5 times larger than an adjacent segment or the luminal irregularity—were not included in the study data.

Recently, there have been research publications that have attempted to predict the occurrence or worsening of coronary artery complications using machine learning methods. Although there have been more studies that have analyzed echocardiographic images [19,20,21], there has also been a report by Azuma J. et al. that has attempted to predict using demographic and laboratory data [22]. Unlike the present study, the model by Azuma J. et al. included the resistance to primary immunoglobulin as a significant feature. The resistance to primary immunoglobulin is a well-known risk factor for the development of coronary artery aneurysm [23,24,25]. This study was designed to predict the development of coronary artery aneurysms using pre-treatment data. Therefore, variables that can only be determined after the completion of primary treatment, such as the resistance to immunoglobulin, total fever duration, and the type of secondary treatment, were not considered as features.

This study attempted to develop a machine-learning-based coronary aneurysm prediction model using a larger number of patients than any other published similar paper to date. Additionally, we expected that, by using machine learning, we could create a predictive model that would outperform statistical techniques, and that the developed model would be used to help determine future intensified primary treatment strategies. However, the performance of the developed model was not satisfactory. A part of the data used in this study was previously utilized in the study by Lam JY et al. for the development of a machine learning model to predict the resistance to primary immunoglobulin [26]. In their study, Lam JY et al. concluded that predictions based on clinical data have inherent limitations and that additional data derived from genomics or proteomics would be necessary to achieve better predictive performance. This conclusion may serve as a valuable reference for interpreting the results of the present study and for planning future improvements. We believe that there are also inherent limitations in using only the demographic and laboratory data included in this study to predict the occurrence of coronary aneurysms, and some additional data from genomics or proteomics would improve the performance of a predictive model. In addition, measurements of coronary artery diameter in the acute phase could not be used as data in this study. Considering the results of Son MBF et al. [8], the inclusion of that measurements could have improved the predictive performance of the model. Further investigations incorporating the coronary artery diameter measured before primary treatment as a feature are warranted.

This study has several limitations. In addition to the acute phase coronary artery diameter measurements mentioned above, several pretreatment variables, such as the time of treatment initiation, were not investigated. In addition, there were a significant number of missing values in the investigated variables, which is also a limitation of the retrospective design. The exclusion of 13,105 patients was mainly due to the absence of coronary artery diameter measurements. However, patients were not excluded solely due to missing measurements of the left circumflex coronary artery. During the early phase of data preparation, we considered that the definition of CAA could be based solely on two major branches—the left anterior descending coronary artery and the right coronary artery—as presented in the study by Friedman et al., which included 2860 patients from North America [27]. Nonetheless, as the analysis progressed, we incorporated measurements of the left circumflex coronary artery into the classification of CAA. We believe that this incorporation may have provided an advantage in identifying patients with coronary artery aneurysms.

In conclusion, a large dataset was utilized to develop a machine learning-based prediction model for coronary artery aneurysms in this study, but its performance was not satisfactory. Future studies should focus on developing improved predictive models by incorporating not only demographic and laboratory clinical data, but also acute-phase coronary artery diameter measurements and/or some experimental biomarkers.

## Figures and Tables

**Figure 1 biomedicines-13-01073-f001:**
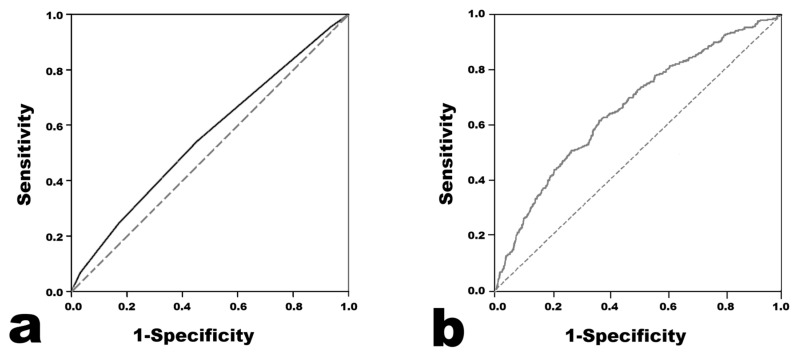
Receiver operating characteristic curves of the models for predicting coronary artery aneurysm. (**a**) Harada score. (**b**) Light Gradient Boosting Machine.

**Figure 2 biomedicines-13-01073-f002:**
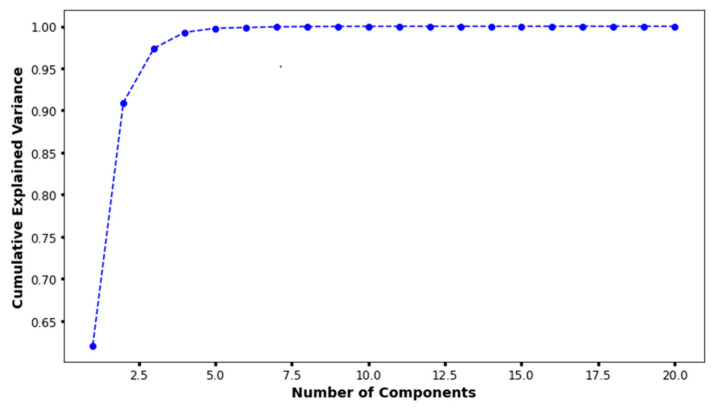
The results of principal component analysis. When there were three principal components, the cumulative explained variance value was 0.974.

**Figure 3 biomedicines-13-01073-f003:**
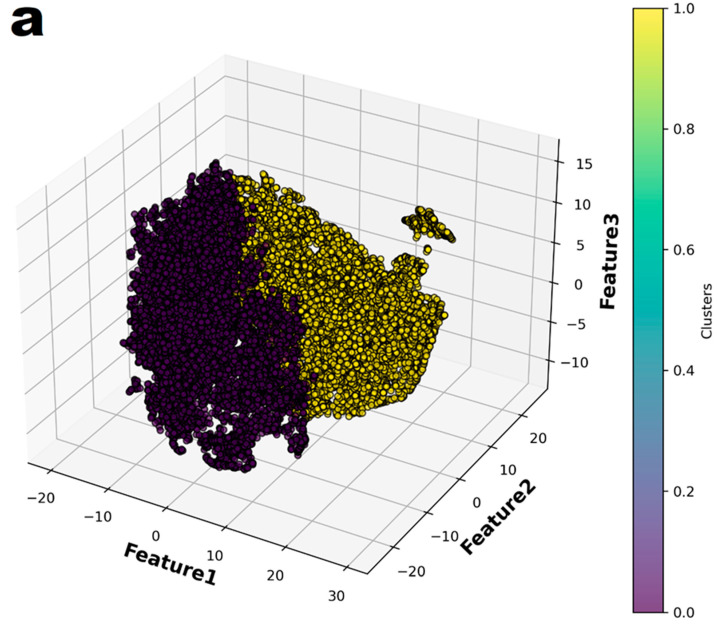
The entire dataset was reduced to three dimensions using t-Distributed Stochastic Neighbor Embedding (t-SNE) and then partitioned into two groups through K-means clustering. (**a**) The two groups were represented in different colors. (**b**) In the entire dataset, the individuals who had coronary artery aneurysms are highlighted in color.

**Table 1 biomedicines-13-01073-t001:** Modified Harada score.

WBC count > 12,000/µL
Hemoglobin < 11.6 g/dL
Platelets < 350,000/µL
C-reactive protein > 3 mg/dL
Albumin < 3.5 g/dL
Age ≤ 12 months
Male sex

**Table 2 biomedicines-13-01073-t002:** Characteristics of the subjects.

Characteristics	Frequency of Sample	Value
Age, months	all	32.9 ± 24.4
Male	all	10,023 (58.3)
Family history	13,008 (75.7)	152 (1.2)
Recurrence	16,537 (96.2)	817 (4.9)
Body weight, kg	all	13.9 ± 5.7
Height, cm	all	91.7 ± 17.1
Body surface area, m^2^	all	0.58 ± 0.17
Complete presentation	all	12,211 (71.0)
Duration of fever, days	16,823 (97.9)	6.2 ± 2.2
Spontaneous defervescence	all	474 (2.8)
Unresponse to initial IVIG	10,000 (58.2)	2260 (22.6)
** *Laboratory findings* **		
WBC, ×10^3^/µL	all	14.1 ± 5.6
Neutrophil, %	all	63.0 ± 16.5
Hemoglobin, g/dL	all	11.4 ± 1.0
Platelet, ×10^3^/µL	all	352.6 ± 115.0
Protein, g/dL	17,037 (99.1)	6.6 ± 1.6
Albumin, g/dL	all	3.9 ± 0.4
AST, IU/L	all	87.1 ± 163.2
ALT, IU/L	all	93.1 ± 148.3
Total bilirubin, mg/dL	16,744 (97.4)	0.68 ± 2.7
Na^+^, mmol/L	16,968 (98.7)	136.5 ± 2.7
C-reactive protein, mg/dL	all	8.0 ± 6.4
Pyuria	16,831 (97.9)	3998 (23.8)
** *Echocardiographic results* **		
LMCA, mm	all	2.50 ± 0.61
Z score	all	0.71 ± 1.45
LAD, mm	all	1.98 ± 0.68
Z score	all	0.58 ± 1.62
LCx, mm	5741 (33.4)	1.68 ± 0.53
Z score		0.31 ± 1.38
RCA, mm	all	2.11 ± 0.67
Z score	all	0.82 ± 1.55
CAA	all	3088 (18.0)

Data are presented as mean ± standard deviation or frequency (%). WBC, white blood cell; AST, aspartate aminotransferase; ALT, alanine aminotransferase; LMCA, left main coronary artery; LAD, left anterior descending; LCx, left circumflex coronary artery; RCA, right coronary artery; CAA, coronary artery aneurysm.

**Table 3 biomedicines-13-01073-t003:** Metrics of performance in each machine learning model.

	AUC (95% CI)	Accuracy	Sensitivity	Specificity	PPV	NPV	F1-Score	MCC	CV
**Logistic Regression (Lasso Regularization)**	0.650 (0.615–0.684)	0.589	0.689	0.567	0.258	0.893	0.376	0.197	0.624
**Logistic Regression (Ridge Regression)**	0.650 (0.617–0.685)	0.589	0.689	0.567	0.259	0.893	0.376	0.197	0.624
**Support Vector Machine**	0.559 (0.521–0.597)	0.659	0.434	0.709	0.246	0.851	0.314	0.117	0.565
**Ensemble Method**	0.641 (0.607–0.676)	0.666	0.534	0.695	0.277	0.872	0.365	0.185	0.636
**Random Forest**	0.621 (0.584–0.654)	0.573	0.625	0.562	0.238	0.872	0.347	0.143	0.617
**Gradient Boosting Machine**	0.655 (0.619–0.687)	0.591	0.676	0.572	0.257	0.890	0.373	0.191	0.637
**Light Gradient Boosting Machine**	0.661 (0.630–0.695)	0.633	0.615	0.637	0.271	0.883	0.376	0.197	0.628
**Multi-Layer Perceptron (3 hidden layers)**	0.545 (0.510–0.584)	0.756	0.217	0.874	0.273	0.836	0.242	0.100	0.539

AUC, area under curve of receiver operating characteristic; CI, confidence interval; PPV, positive predictive value; NPV, negative predictive value; MCC, Matthews Correlation Coefficient; CV, mean ROC AUC from cross validation.

## Data Availability

The original contributions presented in this study are included in the article. Further inquiries can be directed to the corresponding author.

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
