# Peer review of "Limited Performance of Machine Learning Models Developed Based on Demographic and Laboratory Data Obtained Before Primary Treatment to Predict Coronary Aneurysms"

_biomedicines, 2025, doi:10.3390/biomedicines13051073_

Round 1

Reviewer 1 Report

Comments and Suggestions for Authors

The authors aimed to develop a machine learning-based model for coronary artery aneurysm prediction in Kawasaki disease. In particular, although the size of the dataset used in the study is quite good, I think some important points should be explained. These are;

1- Why weren't other performance metrics examined in addition to the ROC curve? Examining metrics such as F1_measure and MCC will benefit the maturation of the article.

2- The sensitivity value of the EBM model is quite low at 0.469. The reason for this should be explained. If there is a limitation related to this situation, it should also be added to the discussion section.
3- Data multiplexing was performed with the Smote technique. However, its situation on the ROC curve should be explained. It should be proven that overfitting did not occur.
4- The performance of the model is considered insufficient for clinical support. Especially the low sensitivity value obtained shows this. The motivation of the study for this very reason should be detailed.
5- What exactly is the innovation that this study provides to science?
6- The limitations of the study were mentioned in a limited way. What kind of preventive measures can be taken for the clinical use of this study?

7- When the publication was examined, it was observed that a sufficient literature review was not done. The authors only reviewed 5 publications in 2024 and 1 publication in 2023. No work published in 2025 was reviewed. It is thought that the authors should expand the literature and pay particular attention to publications in recent years.

Author Response

The authors aimed to develop a machine learning-based model for coronary artery aneurysm prediction in Kawasaki disease. In particular, although the size of the dataset used in the study is quite good, I think some important points should be explained. These are;

  • Why weren't other performance metrics examined in addition to the ROC curve? Examining metrics such as F1_measure and MCC will benefit the maturation of the article.

Response>> I added the F1-score and MCC (Matthews Correlation Coefficient) values ​​to the table. I have added the relevant texts to the manuscript. That is as follows: "The F1 scores were distributed in the low range of 0.242–0.376, and the MCC was distributed in the low range of 0.100–0.197." in 3.1. Supervised machine learning in Results.

2- The sensitivity value of the EBM model is quite low at 0.469. The reason for this should be explained. If there is a limitation related to this situation, it should also be added to the discussion section.

Response>> The EBM model was abandoned, and the related content was removed from the manuscript. This is because, in order to resolve the three points pointed out below, serious overfitting was found through cross-validation analysis, and in the case of the EBM model, overfitting continued even after the SMOTE technique was abandoned. In the case of other models, the overfitting problem was resolved after the SMOTE technique was abandoned.

3- Data multiplexing was performed with the Smote technique. However, its situation on the ROC curve should be explained. It should be proven that overfitting did not occur.

Response>> As included in the answer to the previous question, the SMOTE technique was abandoned due to the overfitting problem. Table 3 was re-created with the results of subsequent reanalysis. The model that showed the highest performance was the light GBM model (ROC AUC 0.661). The model that showed the highest performance when the SMOTE technique was applied was EBM, with an ROC AUC of approximately 0.647, so there was no significant difference. The manuscript was revised according to these changed results. The abandonment of the SMOTE technique was added in 2.3. Supervised machine learning in Materials and Methods as "... (SMOTE) was attempted to generate synthetic samples for the minority class to balance the dataset, but it was ultimately abandoned due to overfitting problems on the training set."

4- The performance of the model is considered insufficient for clinical support. Especially the low sensitivity value obtained shows this. The motivation of the study for this very reason should be detailed.

Response>> We added " Additionally, we expected that by using machine learning, we could create a predictive model that would outperform statistical techniques, and that the developed model would be used to help determine future intensified primary treatment strategies. " in the middle of the Discussion section. This text is the summary of the motivation of the study.

5- What exactly is the innovation that this study provides to science?

Response>> The lesson from this study is that it is difficult to create a satisfactory coronary aneurysm prediction model with conventional clinical (demographic, laboratory) data. This study includes a large number of data, and the fact that machine learning was used to develop the model may also support the lesson above. I have made a major revision to the manuscript as you recommended, so I believe that readers of this manuscript will agree with these thoughts.

6- The limitations of the study were mentioned in a limited way. What kind of preventive measures can be taken for the clinical use of this study?

Response>> I have changed the title of the manuscript to better fit the overall content. The new title is “Limited performance of machine learning models developed based on demographic and laboratory data obtained before primary treatment to predict coronary aneurysms”. I believe that readers who read the entire content will agree with the authors that the prediction model developed in this study is not satisfactory for practical use.

7- When the publication was examined, it was observed that a sufficient literature review was not done. The authors only reviewed 5 publications in 2024 and 1 publication in 2023. No work published in 2025 was reviewed. It is thought that the authors should expand the literature and pay particular attention to publications in recent years.

Response>> Following your recommendation, I searched for publications analyzing Kawasaki disease patient data using machine learning this year. Scientific Reports 2025;15:903 was retrieved, but it was about the development of a prediction model for secondary treatment resistance in Kawasaki disease patients. It was not selected as a reference because it was somewhat distant from this study.

Finally, thank you for your comments and recommendations. Your comments were accurate and useful. I believe that they helped me improve the quality of my paper. I would like to express my gratitude once again.

Reviewer 2 Report

Comments and Suggestions for Authors

The paper is interesting, even though reporting negative results from a clinical point of view.

The most important scientific reason to possibly publish this paper could be then to try to discuss in further detail all the possible reasons why this could happen.   

The abbreviation IVIG should be probably first reported at line 19 and 43.

Line 50: “American”

Lines 78-79: “Although measurements of coronary artery diameter during the acute phase are not included in the dataset”: this could be a critical essential limitation which should be discussed in more detail in the discussion and limitations sections. When the coronary artery diameter measurements were obtained during the course of the disease ?

Lines 84-89: It should be indicated in more detail the reasons why only around 60% of screened subjects could be included . This could be an other reason for the disappointing results from the clinical point of view

Line 110: the number of missing values shoud be reported. This also could be a further reason for  the disappointing results.

In Table 2 it seems that coronary artery measurements were available only in 5,741 cases. So why training and testing the model in 17189 cases (lines 25 and 88) when only a minority had the coronary diameter measurements?   Why not using only these 5,741 cases with coronary diameters available to build up both the training and the testing sets? Possibly the really useful population to be analyzed could not be the one reported at line 88 (17189 cases), but only the 5741 cases with coronary artery diameters available?

Lines 232-236: why not testing also the absolute coronary diameter and the Harada definition and not only the z-score?  

Line 258-259 for the reviewer the main reason for the disappointing results could be, as reported by the Authors:” However, measurements of coronary artery diameter in the acute phase could not be used as data in this study.”

Author Response

The paper is interesting, even though reporting negative results from a clinical point of view.

The most important scientific reason to possibly publish this paper could be then to try to discuss in further detail all the possible reasons why this could happen.   

The abbreviation IVIG should be probably first reported at line 19 and 43.

Response>> Thank you. I added (IVIG) where you pointed it out.

Line 50: “American”

Response>> I changed it to American.

Lines 78-79: “Although measurements of coronary artery diameter during the acute phase are not included in the dataset”: this could be a critical essential limitation which should be discussed in more detail in the discussion and limitations sections. When the coronary artery diameter measurements were obtained during the course of the disease ?

Response>> The data for this study were formed by integrating data from two nationwide surveys that were previously conducted, as mentioned in '2.1 Subjects in Materials and Methods' and 'Institutional Review Board Statement'. The timing of coronary artery measurement is as mentioned in lines 102–103, "In two nationwide surveys, maximum coronary artery diameters were measured in the early course of disease including the subacute phase [14, 15]." In the actual nationwide surveys, data were collected under the above notice. In response to your concern, we have added the following statement, "However, coronary artery diameters measured during the acute phase, including the pretreatment period, were not investigated."

Lines 84-89: It should be indicated in more detail the reasons why only around 60% of screened subjects could be included . This could be an other reason for the disappointing results from the clinical point of view

Response>> Referring to your recommendation, in '2.1 Subjects in Materials and Methods', "A total of 13,105 patients were excluded, for whom coronary artery diameter measurements were incomplete or for whom the Harada score could not be calculated. Patients in whom only the left circumflex coronary artery was missing in the coronary artery diameter measurements were included." was added.

Line 110: the number of missing values shoud be reported. This also could be a further reason for  the disappointing results.

Response>> The 'Frequency of sample' in Table 2 was the number of data that had this value. In cases where all patients had values, it was originally left blank, but with reference to your comment, I added "all".

In Table 2 it seems that coronary artery measurements were available only in 5,741 cases. So why training and testing the model in 17189 cases (lines 25 and 88) when only a minority had the coronary diameter measurements?   Why not using only these 5,741 cases with coronary diameters available to build up both the training and the testing sets? Possibly the really useful population to be analyzed could not be the one reported at line 88 (17189 cases), but only the 5741 cases with coronary artery diameters available?

Response>> The values ​​of the left main coronary artery, left anterior descending artery, and right coronary artery were all present in the study subjects. Subjects with missing values ​​were excluded from the study as mentioned above. However, in the case of the left circumflex coronary artery, even those without value ​​were included in the study subjects. I inserted “The exclusion of 13,105 patients was mainly due to the absence of coronary artery diameter measurements. However, patients were not excluded solely due to missing measurements of the left circumflex coronary artery. During the early phase of data preparation, we considered that the definition of CAA could be based solely on two major branches—the left anterior descending coronary artery and the right coronary artery—as presented in the study by Friedman et al., which included 2,860 patients from North America [27]. Nonetheless, as the analysis progressed, we incorporated measurements of the left circumflex coronary artery into the classification of CAA. We believe that this incorporation may have provided an advantage in identifying patients with coronary artery aneurysms.” Into the limitation paragraph in ‘Discussion’.

Lines 232-236: why not testing also the absolute coronary diameter and the Harada definition and not only the z-score?  

Response>> Regarding the reason why a coronary aneurysm diagnosis was not made according to the Japanese Ministry of Health and Welfare standards, I added a comment like " Further analysis using the original definition of coronary artery aneurysm based on absolute diameter by Japanese Ministry of Health and Welfare [17] —the prediction target of the Harada score—was not performed, as current major guidelines consistently endorse the use of z score–based criteria [1, 18]. And, additional criteria for defining a coronary artery aneurysm - a diameter 1.5 times larger than an adjacent segment or the luminal irregularity - were not included in the study data." to 'Discussion'.

Line 258-259 for the reviewer the main reason for the disappointing results could be, as reported by the Authors:” However, measurements of coronary artery diameter in the acute phase could not be used as data in this study.”

Response>> Good point. I agree with your opinion, and I have repeatedly emphasized the omission of coronary artery diameter measurement in the acute phase in 'Introduction', '2.1 Subjects of Materials and Methods', and the limitation paragraph in 'Discussion'.

Finally, I would like to thank you for your comments which were accurate and helpful. I believe that your recommendations helped me improve the quality of my paper. I would like to express my gratitude once again.

Round 2

Reviewer 1 Report

Comments and Suggestions for Authors

The authors have reflected all changes in the article.

Reviewer 2 Report

Comments and Suggestions for Authors

The Authors made the required changes.